

# Worldwide exploration of the microbiome harbored by the cnidarian model, *Exaiptasia pallida* (Agassiz in Verrill, 1864) indicates a lack of bacterial association specificity at a lower taxonomic rank

Tanya Brown[1], Christopher Otero[1], Alejandro Grajales[2], Estefania Rodriguez[2] and Mauricio Rodriguez-Lanetty[1]

[1] Biological Sciences, Florida International University, Miami, FL, USA
[2] Invertebrate Zoology, American Museum of Natural History, New York, NY, USA

## ABSTRACT

Examination of host-microbe interactions in early diverging metazoans, such as cnidarians, is of great interest from an evolutionary perspective to understand how host-microbial consortia have evolved. To address this problem, we analyzed whether the bacterial community associated with the cosmopolitan and model sea anemone *Exaiptasia pallida* shows specific patterns across worldwide populations ranging from the Caribbean Sea, and the Atlantic and Pacific oceans. By comparing sequences of the V1–V3 hypervariable regions of the bacterial 16S rRNA gene, we revealed that anemones host a complex and diverse microbial community. When examined at the phylum level, bacterial diversity and abundance associated with *E. pallida* are broadly conserved across geographic space with samples, containing largely *Proteobacteria* and *Bacteroides.* However, the species-level makeup within these phyla differs drastically across space suggesting a high-level core microbiome with local adaptation of the constituents. Indeed, no bacterial OTU was ubiquitously found in all anemones samples. We also revealed changes in the microbial community structure after rearing anemone specimens in captivity within a period of four months. Furthermore, the variation in bacterial community assemblages across geographical locations did not correlate with the composition of microalgal *Symbiodinium* symbionts. Our findings contrast with the postulation that cnidarian hosts might actively select and maintain species-specific microbial communities that could have resulted from an intimate co-evolution process. The fact that *E. pallida* is likely an introduced species in most sampled localities suggests that this microbial turnover is a relatively rapid process. Our findings suggest that environmental settings, not host specificity, seem to dictate bacterial community structure associated with this sea anemone. More than maintaining a specific composition of bacterial species some cnidarians associate with a wide range of bacterial species as long as they provide the same physiological benefits towards the maintenance of a healthy host. The examination of the previously uncharacterized bacterial community associated with the cnidarian sea anemone model *E. pallida* is the first global-scale study of its kind.

Corresponding author
Mauricio Rodriguez-Lanetty, rod-mauri@fiu.edu

# INTRODUCTION

Insights into the microbiome diversity of metazoan hosts have triggered a considerable interest in uncovering the regulatory principles underlying host/microbe interactions across multicellular organisms. Over the last several years, microbial symbionts living with vertebrates have been clearly shown to influence disease, physiological and developmental phenotypes in their host (*Blaser et al., 2013*; *Le Chatelier et al., 2013*; *Lozupone et al., 2013*; *Ridaura et al., 2013*). In many marine invertebrates, bacteria associated with host epithelium have also been shown to play a pivotal role in host development (*McFall-Ngai et al., 2013*). For instance, in the bobtail squid, the bioluminescent bacteria, *Allivibrio fisheri* (*Beijerinck, 1889*; *Urbanczyk et al., 2007*) are required symbionts from early host developmental stages so that a functional and healthy light organ can develop (*McFall-Ngai, 1994*; *Nyholm & McFall-Ngai, 2004*). Similar profound effects have been documented for other basal metazoans such as cnidarians. In the case of *Hydra viridis* (Medusozoa: Hydrozoa), induced absence of a microbial community in host polyps causes strong developmental defects and reduces asexual reproduction via budding (*Rahat & Dimentman, 1982*). This suggests that the evolution of microbes and host interactions dates back to earlier diverging metazoan lineages (i.e., cnidarians), which has triggered an imperative interest to understand whether bacterial cores comprised of specific species have evolved in intimate association with their hosts since the early times of metazoan evolution (*Bosch & Miller, 2016*).

Despite the simple body plans in cnidarians molecular analyses of the microbiota associated with these early-diverging organisms, predominantly corals (Anthozoa: Scleractinia), have uncovered an unprecedented bacterial diversity (*Bourne & Munn, 2005*; *Rodriguez-Lanetty et al., 2013*; *Rohwer et al., 2001*; *Sunagawa et al., 2009*). Additionally, the species composition and structure of these microbial partnerships are complex and dynamic. The association between coral host and the consortia of these microorganisms, including bacteria, fungi, viruses and the intracellular microalgae *Symbiodinium*, has been referred to as the coral holobiont (*Rohwer et al., 2001*). Several studies demonstrate that certain bacterial groups associate specifically with some coral species (*Bayer et al., 2013*; *Morrow et al., 2012*; *Rodriguez-Lanetty et al., 2013*; *Rohwer et al., 2001*; *Speck & Donachie, 2012*), implicating the effects of coevolution between coral lineages and certain bacterial strains. However, other studies have revealed that the dominant bacterial genera differ between geographically-spaced hosts of the same coral species (*Klaus et al., 2007*; *Kvennefors et al., 2010*; *Littman, Bourne & Willis, 2010*) and even locally within reefs (*Kvennefors et al., 2010*), which suggests that environmental factors are largely responsible in shaping coral-associated microbial community diversity.

The elucidation of the mechanisms that mediate the complex interactions between microbial communities and anthozoans may be facilitated by studying a tractable model system that can be cultured and manipulated in laboratory conditions. For this purpose, the sea anemone (Anthozoa: Actiniaria) *Exaiptasia pallida* (Agassiz in

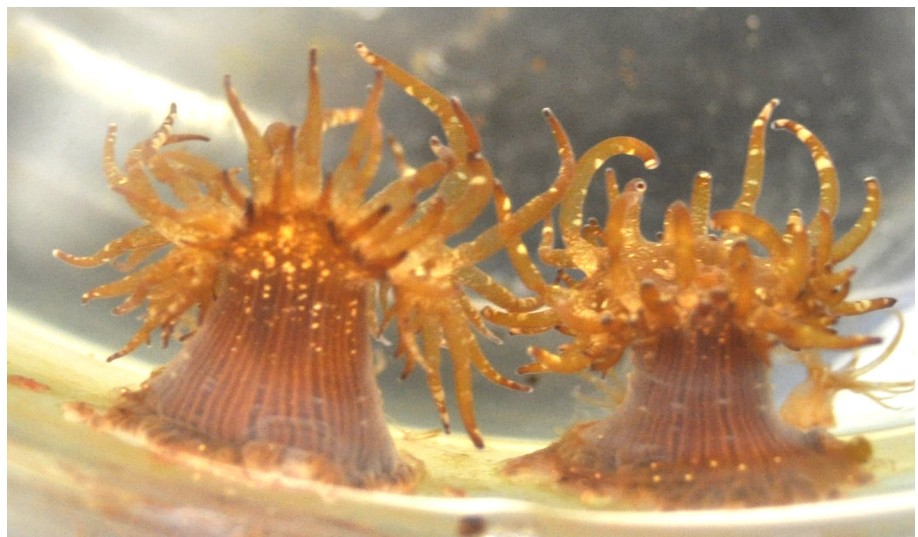

**Figure 1** *Exaiptasia pallida*, the sea anemone species used in the host-associated microbial community study (Photograph taken by Tanya Brown).

*Verrill, 1864*), previously known as *Aiptasia pallida* (see *Grajales & Rodriguez, 2014*), has been proposed as a model organism to study various aspects of the cell biology and physiology of anthozoan-*Symbiodinium* symbiosis (*Weis et al., 2008*) (Fig. 1). The use of this cnidarian model system has advanced our understanding of the molecular and cellular mechanism underlying anthozan/*Symbiodinium* regulation (*Davy, Allemand & Weis, 2012*). Likewise, the *E. pallida* model system could benefit investigations regarding the influence of the associated microbiota on physiological, developmental, and disease-resistant host cnidarian phenotypes. Recent data have shown the importance of the presence of photosymbionts (*Symbiodinium*) on the bacterial community assemblages associated with *E. pallida* (*Roethig et al., 2016*); however, we still lack baseline knowledge about the bacterial diversity and assemblages associated with this sea anemone over wide geographical distributions. In the current study, we characterized the composition, structure and specificity patterns of microbial communities associated with the sea anemone *E. pallida* from worldwide populations using samples from the Caribbean Sea, and the Atlantic and the Pacific oceans. We then compared the microbial composition of natural *versus* specimens reared in the laboratory over different periods of time as means to document the effect of aquarium conditions in the composition and structure of microbial taxa. Furthermore, we analyze the microbial community assemblage variance as a function of the associated-*Symbiodinium* species.

## MATERIAL AND METHODS

### Sample collection, DNA extraction and V1–V3 16S rRNA pyrosequencing

Specimens of *Exaiptasia pallida* were collected from 10 wild populations from different ocean basins worldwide (Table 1; Table S1) including the Caribbean Sea; Northeastern

**Table 1  Location of population sites and sampling information within site.**

| Location ID | Location | Samples (N) | Latitude | Longitude |
|---|---|---|---|---|
| Morelos | Puerto Morelos, Mexico (Caribbean) | 3 | N 20 50 18.57 | W 86 53 02.86 |
| Baja-Sur | Pichilingue, Baja California Sur, Mexico (Pacific) | 4 | N 24 15 46.46 | W 110 36 52.12 |
| Sesoko | Sesoko Island, Okinawa, Japan (Pacific) | 4 | N 26 38 10.70 | E 127 51 55.03 |
| FerryR | Ferry Reach, Bermuda (Atlantic) | 4 | N 32 22 01.51 | W 64 39 36.22 |
| Oahu | Oahu Waikiki, Hawaii, USA (Pacific) | 3 | N 21 16 40.32 | W 157 50 01.04 |
| Florida | Florida Keys National Marine Sanctuary, USA (Atlantic) | 4 | N 25 03 61.03 | W 80 25 38.02 |
| Carenera | Carenera Island, Bocas del Toro, Panama (Caribbean) | 4 | N 09 20 50.34 | W 82 15 18.67 |
| Achotines | Achotines lab, Pedasi, Panama (Pacific) | 4 | N 07 25 50.46 | W 80 11 36.24 |
| Madeira | Madeira Island, Portugal (Atlantic) | 4 | N 32 42 52.84 | W 16 45 47.85 |
| Canaria | Las Palmas Island, Gran Canaria, Spain (Atlantic) | 3 | N 28 19 09.18 | W 15 25 56.89 |
| KML | Outdoor flow-through aquariums at Key Marine Lab, Long Key, Florida, USA | 3 | N 24° 49.567 | W 80° 48.884 |
| Shortlab | Anemones from KML brought to laboratory captivity for 4 Months | 4 | N 24° 49.567 | W 80° 48.884 |
| CC7 | Clone CC7 in laboratory captivity for 6 Years | 3 | Unknown | |
| Petstore | Unknown Collection Site | 2 | Unknown | |

and Western Atlantic; and Eastern, Central and Northwestern Pacific. Ethanol-preserved samples from the populations above were obtained from the Invertebrate collection at the American Museum of Natural History (AMNH). Four additional groups of samples of *E. pallida* reared in captivity were added to the study. One group was obtained from a commercial pet store, a second group from an outdoor flow-through sea water system at the Keys Marine Laboratory (KML, Florida) and the other two were reared in the lab for different time periods: one from a six-year laboratory reared clonal population (CC7) originally obtained from a reef in the upper Florida Keys and the second from anemones more recently collected from the KML (Florida) and maintained in the lab for four months. Anemones maintained in aquaria were fed twice a week, and were sampled two days after their last feeding for this study. The anemones were rinsed with sterile seawater before freezing them in liquid nitrogen. Total DNA was extracted from the entire body of the collected sea anemone samples (3-4 per population site) using the DNeasy Plant Mini Kit DNA (Promega, Madison, WI) following the standard protocol recommended by the manufacturer.

To assess DNA quality and lack of PCR inhibition, the universal bacterial primers 27F (5′-AGAGTTTGATCMTGGCTCAG-3′) and 1492R (5′-TACGGYTACCTTACGACTT-3′) (*Jiang et al., 2006*) were used to amplify a region of nearly 1500 bp following our previously published protocol (*Rodriguez-Lanetty et al., 2013*). Total DNA from samples in which 16S was successfully amplified was sent to the sequencing facility at the Molecular Research LP (Shallowater, Texas, USA). Barcoded 16S amplicon sequencing was performed using the trademark service bTEFAP® as described by *Dowd et al. (2008)*. The 16S universal eubacterial primers used were: ill27Fmod 5′-AGRGTTTGATCMTGGCTCAG-3′ and ill519Rmod 5′-GTNTTACNGCGGCKGCTG-3′. A single-step 30 cycle PCR using HotStarTaq Plus Master Mix Kit (Qiagen, Valencia, CA, USA) was used under the following

conditions: 94 °C for 3 min, followed by 28 cycles of 94 °C for 30 s; 53 °C for 40 s and 72 °C for 1 min; subsequently, a final elongation step at 72 °C for 5 min was performed. Following the PCR, all amplicon products from different samples were mixed in equal concentrations and purified using Agencourt Ampure beads (Agencourt Bioscience Corporation, Beverly, MA, USA). Prepared 16S library samples were sequenced utilizing Roche 454 FLX titanium instruments and reagents following manufacturer's guidelines with the goal of examining the diversity (richness and abundance) of bacterial species associated on the surface and within the tissue of *Exaiptasia pallida*. The high throughput sequencing was done on the V1–V3 16S rRNA region (∼495bp).

## Analysis of microbial community

The Q25 sequence data derived from the sequencing process was processed using the MR DNA ribosomal and functional gene analysis pipeline (www.mrdnalab.com, MR DNA, Shallowater, TX, USA). Sequences were depleted of barcodes and primers; short sequences <150 bp were also removed. Sequences with ambiguous base calls and homopolymer runs exceeding 6 bp were also removed. Sequences were then denoised and chimeras removed using UCHIME (*Edgar et al., 2011*; *Legendre & Gallagher, 2001*). Operational taxonomic units (OTU) were defined clustering at 3% divergence (97% similarity) and taxonomically classified using BLASTn against a curated database derived from RDP (http://rdp.cme.msu.edu; *DeSantis et al., 2006*) and NCBI (http://www.ncbi.nlm.nih.gov) implemented through an analysis pipeline developed by Molecular Research DNA. A consensus sequence from each OTU was determined by majority and used for the taxonomic classification (from phylum to the species level). The sequence dataset has been archived in NCBI (Accession numbers: SAMN06130328 to SAMN06130380).

The species richness estimator Chao1, and the Shannon–Wiener index (H') of diversity were calculated to evaluate the expected number of unseen species and the level of alpha-diversity across the samples, respectively. To assess the level of diversity detected as a function of sequencing effort rarefaction analyses were also carried out. One-way analysis of variance and post hoc Tukey's HDS comparisons were conducted to test for significant differences in alpha diversity across the studied sites. Community similarity analysis was performed by nonmetric multidimensional scaling (nMDS) using the Bray–Curtis distance metric after Hellinger standardization (*Legendre & Gallagher, 2001*). This analysis was conducted in the R version 3.02 package VEGAN (*Core, 2011*; *Oksanen et al., 2011*). Furthermore, spatial patterns in community composition and structure were explored using hierarchical cluster analysis in PRIMER-E (*Clarke & Warwick, 2001*). A permutation similarity profile test (SIMPROF; (*Clarke, Somerfield & Gorley, 2008*)) was performed to identify clusters of samples with statistically significant internal structure ($p < 0.05$). The number of permutations performed was 999 and the resemblance measure used was S17 Bray-Curtis similarity.

## Symbiodinium composition hosted by *Exaiptasia pallida* anemones

To determine whether the patterns of bacterial assemblages associated with *Exaiptasia pallida* across the different geographical sites was correlated with differential composition

of associated *Symbiodinium* we conducted species genetic identification of the microalgal symbionts. The *Symbiodinium* identity in anemones from 10 out of the 14 sampling locations were obtained from metadata published by our group in *Grajales & Rodriguez (2016)*. *Symbiodinium* identification from anemones of the other four studied captive populations (Petstore, KML, shortlab, and CC7) were conducted for this study using the chloroplast ribosomal 23S hypervariable region (cp23S-HVR) as described by *Granados-Cifuentes et al. (2015)*. This region was amplified using the forward primer 23SHYPERUP (5′-TCAGTACAAATATGCTG-3′) (*Santos, Gutierrez-Rodriguez & Coffroth, 2003*) and reverse primer 23SHYPERDN (5′-TTATCGCCCCAATTAAACAGT-3′) (*Manning & Gates, 2008*). The PCR reactions consisted of a final volume of 20 µl using GoTaq Green Master Mix (Promega, Madison, WI) that was adjusted to 25 nM $MgCl_2$ and a primer concentration of 0.4 µM. The PCR profile consisted of an initial denaturation cycle of 94 °C for 2 min, followed by 42 cycles at 94 °C for 20 s, 50 °C for 30 s, and 72 °C for 30 s, and a final extension at 72 °C for 10 min. Cleaned PCR products were Sanger sequenced in the in-house DNA core facility at Florida International University.

## RESULTS

Sequencing efforts produced a total 679,061 reads for the V1–V3 16S rRNA from 49 anemone samples (3–4 replicates per population site) and an average of 13,581 reads per samples. After quality-based filtering a total of 507,882 reads were obtained representing an average of 10,157 reads per sample. To assess differences in community assemblages across samples sequences were clustered into operational taxonomic units (OTUs), based on a similarity level of 97%. Rarefaction analyses suggested that the sequencing effort per population site seemed sufficient for the estimation of OTU diversity for most of the sampled sites; however, the maximum expected species richness was not reached in most of the studied sites (Fig. S1).

A high richness of bacterial species (a total of 12,585 OTUs) was revealed to engage in association with the sea anemones across ocean basins (Table S1). The highest average bacterial OTU richness were obtained from the *E. pallida* samples obtained from a commercial pet store ($1,671 \pm 144$) and the CC7 clonal population samples reared in the lab ($1,358 \pm 225$), which were approximately four and three times higher respectively than anemones with the lowest OTU richness from a natural population in Hawaii ($409 \pm 227$). Interestingly, no significant differences in alpha diversity based on Shannon-Wiener index were detected across populations (One-way ANOVA and Tukey's HDS $p > 0.1$, Table S1).

Classification of sequences on the phylum level revealed that bacterial communities were dominated by *Proteobacteria* and *Bacteroides* followed by the phyla *Firmicutes* and *Cyanobacteria* (Fig. 2). *Proteobacteria* dominated the communities ($>50\%$) associated with samples collected from all wild populations (Pacific and Atlantic oceans, Caribbean Sea). However, the bacterial communities associated with some anemones from the Northeastern Atlantic and from an outdoor flow-through sea water system in KML (Florida Keys) were dominated by *Firmicutes* (nearly 70%; Phyla Indicator Analysis, $p < 0.001$). Besides these four major represented phyla across samples, most of the other rare phyla were not

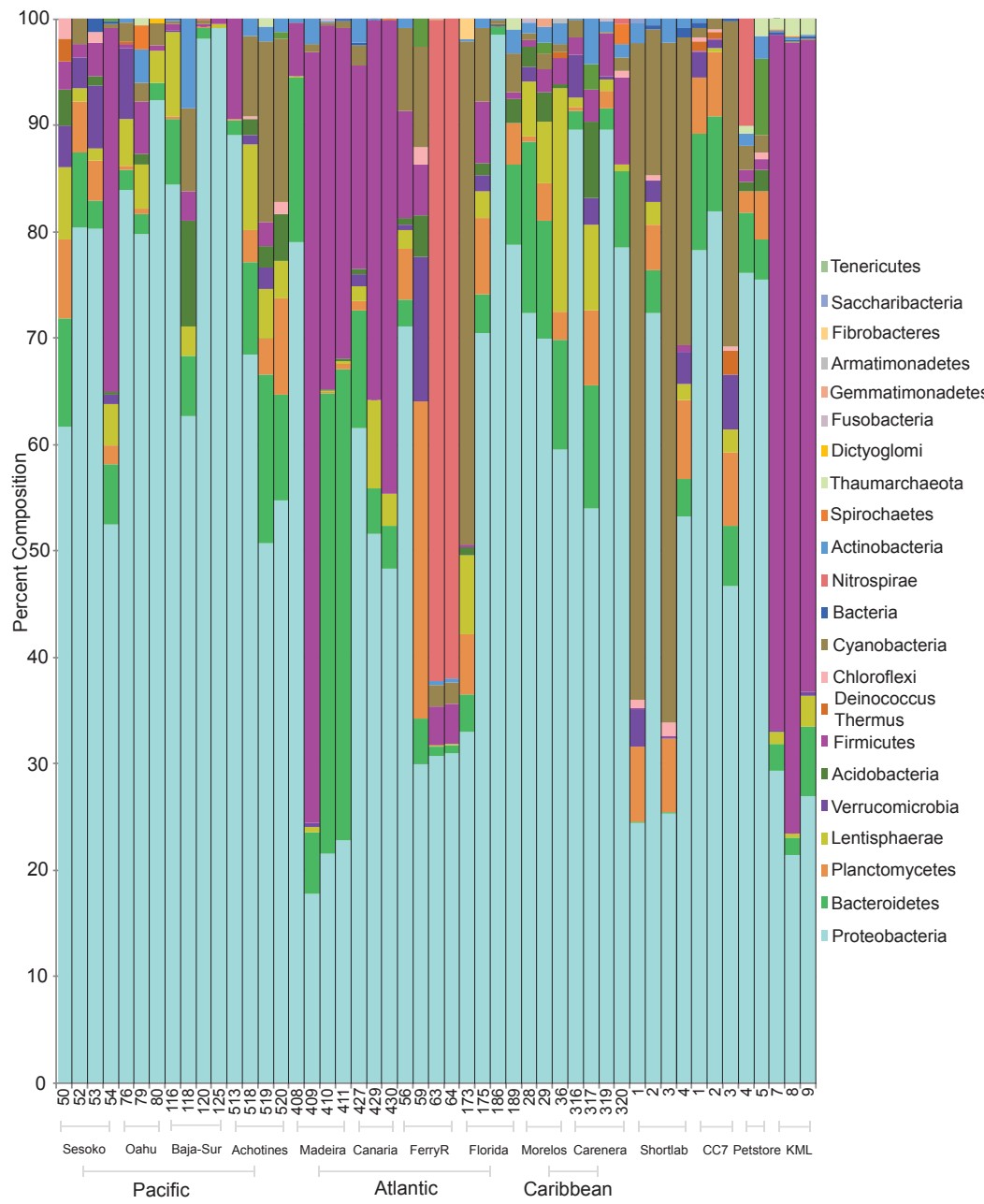

**Figure 2** Prevalence and distribution of bacterial phyla identified in the microbial community associated with the anemone *Exaiptasia pallida* across all natural population sites, including the lab reared populations.

indicators of a specific association with a particular geographical location (Phyla Indicator Analyses, $p > 0.05$).

Anemones collected from the KML outdoor sea water system were transported and maintained in re-circulating indoor aquaria using artificial seawater during a period of four months (population referred to as shortlab). During this short period the microbial community underwent a considerable shift of bacterial phyla. *Firmicutes* decreased from

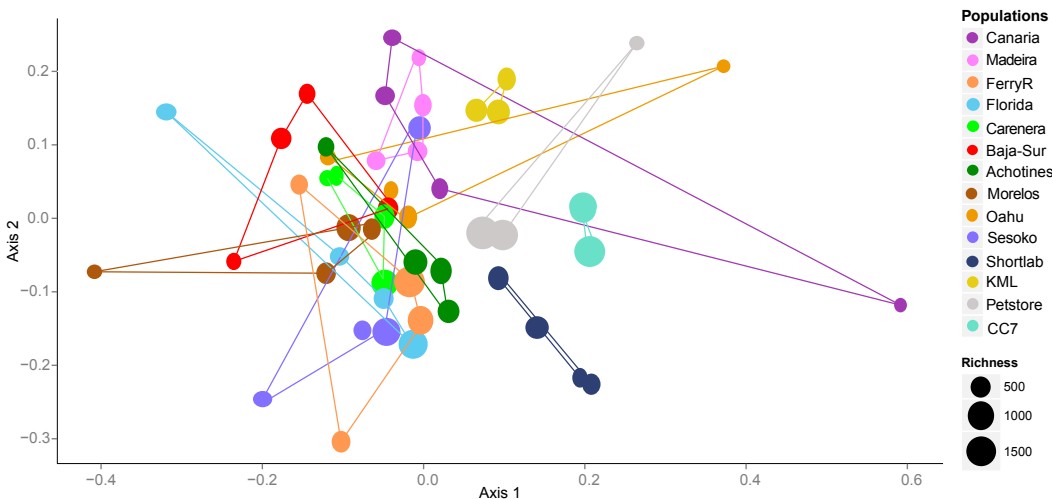

**Figure 3** **Nonmetric multidimensional scaling (nMDS) ordination using V1–V3 16S rDNA OTUs (derived from high throughput 454 sequencing) of the microbial community associated with *Exaiptasia pallida* from all collected natural and captive populations.** nMDS are based on Bray–Curtis dissimilarity distance after Hellinger transformation and Kruskal's stress is 0.206. Richness of OTUs per anemone sample in each population site is proportional to the size of the data sample point on the graph.

~70% to abundances of less than 1%, and *Cyanobacteria* and *Proteobacteria* increased in relative abundances, ~42% and 44% respectively. The bacterial communities associated with anemones reared in the laboratory for six years (clone CC7, from an unknown Florida Keys population), using Instant Ocean Water, were dominated by *Proteobacteria* (68%), with similar relative abundances to those detected in many other wild populations.

Within *Proteobacteria*, the class *Alphaproteobacteria* was most commonly dominant in anemones from the North Pacific (51%), Caribbean (53%), Atlantic (47%), four-month lab reared/shortlab (58%), and six-year lab reared/CC7 (78%) populations (Fig. S2). *Gammaproteobacteria* and *Alphaproteobacteria* were equally common in the Eastern Pacific (44% and 40%) and the commercial pet store (48% and 44%) anemone populations. *Gammaproteobacteria* was found to be most common in Central Pacific anemone samples (53%). The anemones from the outdoor sea water system at the KML that were dominated by *Firmicutes* also had a very distinct group of *Proteobacteria*, represented mainly by *Deltaproteobacteria* (45%).

By examining the composition and structure of the bacterial community associated with *E. pallida* based on OTUs, we detected considerable differences among populations and geographical locations. The multivariate ordination of the bacterial communities did not exhibit clear grouping of the samples based on geographical origin (nMDS, Fig. 3). However, anemone samples reared in captivity showed less variability and each of the three captivity groups (4-month *versus* 6-year *versus* pet store) clustered in its own ordination grouping.

Hierarchical cluster analyses and similarity profile test (SIMPROF) were performed to detect bacterial community structure among the samples independent of their geographical origin. These statistical analyses revealed that very distinct microbial communities

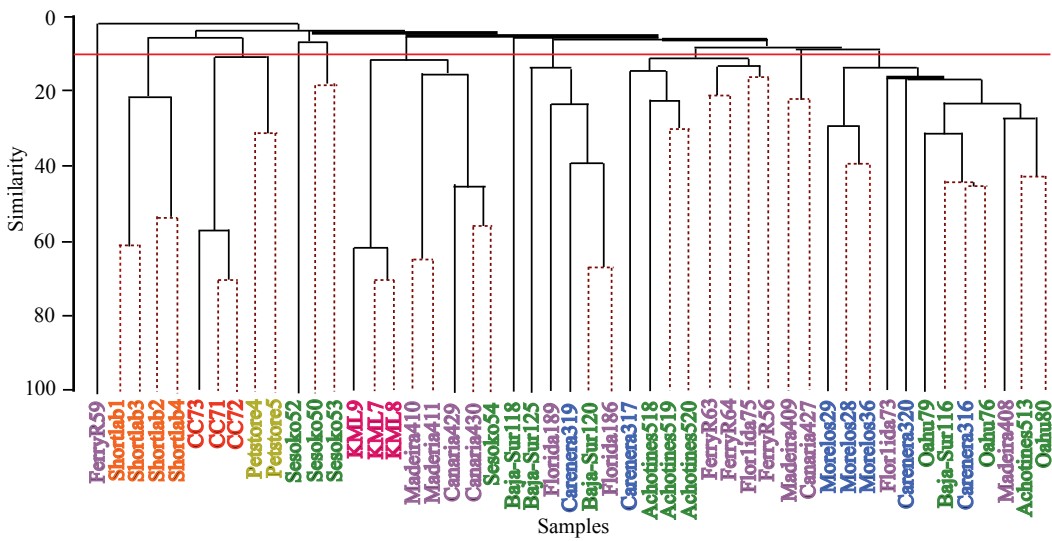

**Figure 4  Hierarchical clustering dendogram of bacterial communities associated with *Exaiptasia pallida* specimens from all wild and captive populations.** Solid lines indicate significant branches (SIMPROF, $p < 0.05$) while dashed lines are unsupported. Red line indicates location at which populations were clustered during the SIMPROF analysis. Colors of samples indicate geographical sampling location: Blue—Caribbean, Purple—North Atlantic, Green—Pacific, Pink—Keys Marine Lab, Yellow—Commercial Pet Store, Orange—4 Month Captive (shortlab), Red—6 Year Clonal Captive (CC7). The numbers indicated below the sample names display the significant SIMPROF groupings.

characterized most of the samples. While the 49 anemone specimens were collected from ten and four wild and captive populations respectively, SIMPROF analyses detected 32 significant bacterial assemblages (SIMPROF, $p < 0.05$; Fig. 4). Out of these 32 groupings, fifteen groups were conformed by two samples and only one group was conformed by three samples. The remaining 16 samples were not clustered in any group indicating their unique bacterial assemblage. These clustering analyses produced similar results to the previous nMDS analysis and revealed that the associated bacterial community in wild anemones did not show geographical patterns. On the other hand, bacterial communities from captive *E. pallida* were more similar to each other. Out of 12,585 OTUs, no single bacterial OTU was shared among all anemones regardless of the geographic origin. The most prevalent OTU (OTU9148: *Vibrio tubiashii*) was found in 75% of the samples and only 92 OTUs (less than 1% of the total discovered OTUs) were shared by one quarter of all samples (Fig. S3). Based on taxonomic classification, species within the genera *Vibrio*, *Nautella*, *Ruegeria*, *Marinobacter*, *Lentisphaera*, and *Flaviobacterium* were common representatives within the microbial community associated with *E. pallida*.

Regarding the *Symbiodinium* composition associated with *E. pallida*, we detected five *Symbiodinium* types associated with the studied *Exaiptasia* samples (Table S2). The majority of the anemones hosted single symbiont species represented mostly by *Symbiodinium* B1. Population sites, including the Florida KML populations, Puerto Morelos (Mexico) and the CC7 indoor aquarium population, hosted *Symbiodinium* A4. Two populations, including Bermuda and another Florida population, hosted a mixture of *Symbiodinium* species composed of B1/C1 and A4/B2, respectively. Interestingly, the two anemones obtained

from the local pet store each hosted a different *Symbiodinium* type (C3 andA4). Comparison of bacterial community and *Symbiodinium* composition is discussed below.

## DISCUSSION

This study revealed the previously uncharacterized bacterial community associated with the cnidarian sea anemone model *Exaiptasia pallida* at different locations throughout the Northern Hemisphere. This examination of anemone/bacterial association is the first global-scale study of its kind. The results show that a complex and diverse microbial community colonizes the anemone and varies considerably both among and within sampling locations. This indicates a lack of a bacterial core community evolving in intimate association with this cnidarian host. The pattern of community assemblages across sampling locations and geographies does not correlate with the composition of *Symbiodinium* symbionts associated with the anemones (Table S2). Anemones from seven populations across different oceans (including North and Central Pacific, Caribbean and Atlantic) harbored the same genetic species of *Symbiodinium* B1 (*Grajales, Rodríguez & Thornhill, 2016*; Table S2); however, their microbiomes were different as indicated by the multivariate test analyses (Figs. 3 and 4). Furthermore, in some cases anemones from same sampling sites depicted very similar microbiomes but showed to harbor distinct *Symbiodinum* composition. For instance, two anemones collected from the local petstore each hosted different *Symbiodinium* species (C3 and A4), however their microbial communities were closer to each other than microbiomes from other sampling sites. This indicates the lack of specific bacterial species associated to a specific photosymbiont type, at least at the level of resolution our study explored. It also highlights the fact that the anemone does not exert high selectivity in shaping the associated microbial community. Although it has been shown that coral reef invertebrate microbiomes correlate with the presence of photosymbionts (*Bourne et al., 2013*; *Roethig et al., 2016*), our data indicate that the type/species of photosymbiont (in such case *Symbiodinium*) does not seem to explain natural variability observed in the microbial community associated with *E. pallida* across distinct geographical locations. It is expected that the metabolic contribution from *Symbiodinium* to the anemone host has an effect structuring the associated bacterial community compared with aposymbiotic anemones lacking *Symbiodinium*, but differential *Symbiodinium* species composition may not drive further changes of anemone microbiomes.

When examined at the phylum level, bacterial diversity and abundance associated with the cosmopolitan sea anemone *E. pallida* are broadly conserved across geographic localities with samples containing largely *Proteobacteria* and *Bacteroides.* These two phyla have been documented to represent the most abundant bacteria associated with scleractinian coral, a close relative of sea anemones (*Ainsworth et al., 2015*; *Chu & Vollmer, 2016*; *Rodriguez-Lanetty et al., 2013*; *Rohwer et al., 2002*; *Sunagawa et al., 2009*). However, the species-level makeup within these phyla differs drastically across space suggesting a high-level core microbiome with local adaptation of the constituents. There was no a single bacterial OTU ubiquitously found in all anemones samples. This finding differs from the postulation,

based on a study conducted in the class Hydrozoa (*Fraune & Bosch, 2007*), that cnidarian hosts actively select, regardless of environmental conditions, and maintain species-specific microbial communities (*Fraune & Bosch, 2007*). Our results indicate that differences in global and local environmental factors might play important roles sorting the composition of bacterial species that associates with the anemone *E. pallida*. This paradigm is supported by a recent study which also demonstrated that environmental parameters such as salinity, dissolved oxygen, and ammonium are key drivers in the regulation of the composition and structure of bacterial communities associated with scleractinian corals (*Lee et al., 2012*). Moreover, changes in the microbial community structure were revealed after rearing specimens of *E. pallida* in captivity within a period of just four months, supporting the idea that differences in aquatic environments have a strong effect on shaping associated bacterial assemblages at the species level.

The finding of *Vibrio tubiashii* as the most ubiquitous bacteria associated with *Exaiptasia pallida* (found in 75% of all samples) was interesting, as this bacterium was recognized as a significant pathogen of several species of bivalve larvae in the 60's (*Tubiash, Colwell & Sakazaki, 1970*). The bacteria was also shown to re-emerge and cause vibriosis in shellfish hatcheries on the west coast of North America causing decline in larval oyster production of up to ∼59% in one hatchery (*Elston et al., 2008*). It has also been demonstrated that strains of this species can also cause disease in massive corals in the Indian Ocean (*Sere et al., 2015*). Although the functional and ecological significance of the association between this bacteria species and *Exaiptasia pallida* remains to be discovered, our findings indicate that these widely distributed anemones could be reservoirs of the pathogen *Vibrio tubiashii*. Potential evidence supporting this hypothesis is the interesting fact that oyster farms have been one of the plausible means responsible for the spread of *E. pallida* worldwide (*Thornhill et al., 2013*), due to the close contact between bivalves and sea anemones. This association has also been observed in different natural environments, such as mangrove roots, where *E. pallida* is often found growing on top of different bivalves (A Grajales, E Rodriguez & M Rodriguez-Lanetty, pers. obs., 2016). However, the mode of transmission of the pathogenic bacteria from the anemones to susceptible host organisms requires investigation.

Unlike in hydrozoans, our findings suggest a lack of coevolution between a sister lineage within Cnidaria (Anthozoa: Actiniaria) and specific bacteria. Within anthozoans, most of the studies exploring the bacterial diversity via culture-independent approaches have been done within the subclass Hexacorallia, more specifically within the order Scleractinia (i.e., stony corals). In this group a number of coral-associated microbial exploratory studies have shown that corals harbor some of the most highly diverse and abundant microbial communities in marine invertebrates after Porifera (*Bourne & Webster, 2013*; *Mouchka, Hewson & Harvell, 2010*; *Rodriguez-Lanetty et al., 2013*; *Sunagawa et al., 2009*). Evidence supporting a clear co-evolution or co-diversification pattern between prokaryotes and corals is however absent. While some studies have shown species-specific patterns of bacteria/host associations (*Littman et al., 2009*; *Sunagawa, Woodley & Medina, 2010*), recent studies using high throughput 16S rRNA gene sequencing have shown that microbial communities associated with scleractinian corals are not species specific (*Hester et al.,*

*2016*; *Meistertzheim et al., 2016*; *Zhang et al., 2015*) and are controlled primarily by external environmental conditions rather than the coral holobiont (*Pantos et al., 2015*). Although there may be little support of co-evolutionary patterns, it seems that some core bacteria groups might have broadly specialized to associate with scleractinian corals regardless the host species lineage (*Ainsworth et al., 2015*).

Based on our findings, we propose that more than maintaining a specific species composition of bacteria, *E. pallida*, and perhaps many other anthozoans, associate with a wide range of bacterial species as long as they provide the same physiological benefits towards the maintenance of a healthy host. To certain extent this explanation is supported by the fact that at higher taxonomic level we detected more similarities across populations in the host-associated microbial structure. The particular bacterial assemblage that may engage in symbiosis with the anemone host will then depend on the existing pool of bacterial species filtered by the environmental conditions of the host habitat, provided that these bacteria belong to a preferred bacterial group with similar ecological functions. It is interesting to note that current global distribution of *E. pallida* seems the result of recent invasion events based on the lack of host population genetic structure (*Grajales & Rodriguez, 2016*; *Thornhill et al., 2013*), and yet we were able to detect a complete turnover of the bacterial community at the species level associated with this invasive host anemone across global scale. Our study highlights a potential role of the environment to delineate the patterns of host/microbial symbiont associations.

## ACKNOWLEDGEMENTS

We would like to thank Ms. Cindy Lewis for collecting sea anemones in the Florida Reef Tract and Dr. John Pringle for providing the CC7 *Exaiptasia pallida* clonal line used in this study. We are grateful to the member of the IMaGeS Lab, Mr. Daniel Merselis, Dr. Anthony Bellantuono, Ms. Katherine Dougan, Ms. Cindy Lewis, and Ms. Ellen Dow for their comments during the writing of early draft versions of this manuscript. Comments of three anonymous reviews improved this manuscript.

### Funding

Mauricio Rodriguez-Lanetty received funds from the National Science Foundation (NSF-IOS-1453519) to conduct this research. Estefania Rodriguez and Alejandro Grajales received funds from the Lerner-Gray Fund for Marine Research and a NSF Doctoral Dissertation Improvement Grant (NSF DEB 1110754). The funders had no role in study design, data collection and analysis, decision to publish, or preparation of the manuscript.

### Grant Disclosures

The following grant information was disclosed by the authors:
National Science Foundation: NSF-IOS-1453519.
NSF Doctoral Dissertation Improvement: DEB 1110754.

## Competing Interests

Mauricio Rodriguez-Lanetty is an Academic Editor for PeerJ.

## Author Contributions

- Tanya Brown performed the experiments, analyzed the data, contributed reagents/materials/analysis tools, wrote the paper, prepared figures and/or tables, reviewed drafts of the paper.
- Christopher Otero performed the experiments, analyzed the data, contributed reagents/materials/analysis tools, reviewed drafts of the paper.
- Alejandro Grajales and Estefania Rodriguez performed the experiments, contributed reagents/materials/analysis tools, reviewed drafts of the paper.
- Mauricio Rodriguez-Lanetty conceived and designed the experiments, performed the experiments, analyzed the data, contributed reagents/materials/analysis tools, wrote the paper, prepared figures and/or tables, reviewed drafts of the paper.

## DNA Deposition

The following information was supplied regarding the deposition of DNA sequences:

The sequence dataset has been archived in NCBI (Accession numbers: SAMN06130328–SAMN06130380).

## Data Availability

Data from: Worldwide exploration of the microbiome harbored by the cnidarian model, *Exaiptasia pallida* (Agassiz in Verrill, 1864) indicates a lack of bacterial association specificity at a lower taxonomic rank. Dryad Digital Repository. doi: 10.5061/dryad.05st3.

## Supplemental Information

Supplemental information for this article can be found online at http://dx.doi.org/10.7717/peerj.3235#supplemental-information.

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
