# Peer review of "Worldwide exploration of the microbiome harbored by the cnidarian model, Exaiptasia pallida (Agassiz in Verrill, 1864) indicates a lack of bacterial association specificity at a lower taxonomic rank"

_PeerJ, doi:10.7717/peerj.3235_

## Round 0.1 · original submission · Major Revisions

I have heard back from three reviewers, all of whom were positive about your manuscript. However, all three have various suggestions on how to improve your work. I have looked over the manuscript and these comments, and find them to be helpful and constructive. In particular, please carefully consider your responses to reviewer 1's comments on prevalent OTUs, reviewer 2's comments on SIMPROF. taxonomic names, and Symbiodinium information, and reviewer 3's request for more information in the Materials and Methods.

As the comments cover a wide range of areas and your responses to these will take some time and work, my decision is 'Major revision'.

·

Basic reporting

Brown, et al., Describe the bacterial diversity (microbiome) associated with multiple samples of the cnidarian Exaiptasia pallida taken from diverse geographical locations. The structure and language used in the paper are clear and understandable. Their introduction clearly positions the lack of information among microbial/host interactions on this Genus. They present relevant information of literature related to other cnidaria, which highlights the need to study the microbial community associated with a sea anemone and they emphasize the use of a cultivable model that can be manipulated in the laboratory for further studies.
All figures (including supplementary) are relevant to the discussion of their results.
I could not find, however, any mention of the sharing of their data.

Experimental design

Their experimental design serves the purpose of the hypothesis by selecting specimens from diverse locations, including some from laboratories and a pet shop which sums up most of the environments where the sea anemone can be found.
The methods are what is considered state of the art regarding microbiome description. They use the correct statistical tools to analyze the data.

Validity of the findings

Although the authors discuss their data according to their findings, I believe that a lower taxonomic analysis discussion should be included. This would provide even more support to their conclusions.
First thing I suggest is discussing the diversity of phyla present when the two most prominent ones are removed (i.e. without Proteobacteria and Bacteroidetes/Firmicutes), and I would discuss more deeply the ecological role of bacteria belonging to these phyla and if they have been found in association with other anthozoa.
Although they mention that there were OTUs that were dominant (V. tiubiashii), they do not discuss on what the role of this genus and the other 1% that was shared could have in the host.
The discussion is centered upon highlighting that there is no common core among the organisms, which is highly evident and important, of course, but lacks the robustness of discussing what is found in the host "per se" and very importantly, emphasizing also what makes them different. For example, the statistic analysis that shows no geographical patterns does show some outliers for each group, what makes these outliers so different to the rest of their group?
I believe including the discussion of this data would improve the manuscript and make it much more than a mere description of the OTUs that are present to say there is nothing in common.

Additional comments

I think this paper is quite important to the field in order to support the idea that microbiomes are not only determined by the host itself but by the whole environment in which they are in. However, I believe the data can be discussed much better by viewing not only what is similar, but also what makes the differences.
Also, including a discussion on the most prevalent OTUs should be included. I see papers leaving all discussions at the phylum level since the genus one is so different between their samples, I think this should be part of the analyses and discussion on it should always be encouraged so that we can understand microbial-host associations much better.
Please include the reference to the raw data depository.

Reviewer 2 ·

Basic reporting

In figure 2 and 3, there are colors very similar which it is hard to distinguish, can author try to improve it?

Experimental design

Please provide the sequencing device for high throughput sequencing.

Authors listed out three taxonomic classify databases, some of the taxonomical names are not consistent among three databases. Please state more clearly on the steps and parameters for taxonomic assignment.

Please also list the parameters for SIMPROF.

I suggest authors can provide the host genetic information. Although all the samples might be considered as same species, Thornhill et al. 2013 did find two genetic lineages. Symbiodinium might also play an important role on shaping microbial community structure. Symbiodinium lineages associated with E. pallida might easily change under the tank environment. It might be worthy to survey the Symbiodinium lineages for this research.

Validity of the findings

Can authors provide the rarefaction curve to show the sufficiency of sequencing depth?

As mentioned in the experimental design, if authors think there is no need to provide the information on host and Symbiodinium, please elucidate in the discussion.

Additional comments

No Comments

Reviewer 3 ·

Basic reporting

The article is written in good English and well structured. However, the materials and methods section is incomplete:

The samples used are not described in enough detail. The 10 wild populations are from museum specimens, however sampling dates are not mentioned in Table 1. It could be possible that various different bacterial species differ in stability in ethanol storage (different cell wall types etc.). Or maybe bacteria externally attached to the specimen get lost in the ethanol storage faster than those in the body cavity/tissues. For these reasons it would be good to know the time in storage for the specimens and if it differs among them.
If available it would also be nice to know how closely to each other the individuals of each population were sampled, i.e. if they are likely to be clones.
Also, was the health status of the samples assessed in any way? Were they all of similar size/age?

How were the aquarium specimens sampled? Were they rinsed with sterile sea water? What was the feeding status at sampling time, if they were being fed at all? Is it know if the individuals are from a clonal population?

Line 121 f.: “Total DNA … were then pyrosequenced…” I don’t understand this part unless it is only a typo. Surely the V1-V4 PCR products were sequenced?

Line 125: This needs more details. Which primers were used? Which PCR conditions, polymerase? PCRs in triplicate and then pooled? Cleanup steps? The PCR in 16S tag sequencing experiments is a crucial step and deserves to be described with more detail.
How were the primers barcoded for pooled sequencing? Was there a linker sequence? Which sequencing machine and chemistry was used, Roche 454 FLX?

Where are the raw data available? This is a requirement, see the PeerJ author instructions.

The sequence processing and resulting dataset is also described too briefly. How many raw sequences were there, in total and per sample? How many were filtered out? Was error correction used? Were the sequences quality trimmed? Which clustering algorithm was used, was there an alignment and if yes to which dataset? Why were sequences clustered first and only then chimeras removed? Clustering with chimeras removed may yield different clusters. Were there any contaminating sequences from dinoflagellate symbiont chloroplast 16S genes? How were those removed?

Line 132ff: How was BLASTn used for OTU classification? Was only the top hit used? Was there an e-value cutoff, or one for query coverage? Was a representative sequence from each OTU used or were all blasted? If the former, how was the representative sequence determined? If the latter, how was the consensus taxonomy for an OTU decided?
How was this database derived from the ones named, and in what way was it curated? The source databases use different taxonomies, how are those merged in your database?

The methods only state classification was performed on the OTU level. What data exactly went into figure 2, is the percent value percent of all OTUs or percent of all sequences? Generally these stacked bar plots should represent individually classified sequences independent of OTUs to represent bacterial abundance in the samples.

For the statistical analyses, was the OTU data subsampled or normalized among samples?

More data should be added to Suppl. Table 1, it is useful to calculate values for diversity and estimated richness, such as the ACE or Chao1 index and Simpson diversity. These allow comparison to other organisms.

Experimental design

The authors do not present 16S diversity data for any water samples. Generally I would agree this is not strictly necessary anymore, as all kinds of previous studies show that seawater is very different from host (such as coral) tissue in its microbial community. Also it is obviously not possible to sequence surrounding water for the museum specimens anymore. However especially in the case of samples moved from the KML outdoor tanks to the indoor aquaria I would consider the water samples an important control, because of the strong shift in microbiome makeup. With the data presented we can’t be sure what part of this shift is actually caused by a differing water microbiome. In this regard it is also important to know who the samples were taken, as mentioned above. See also line 246 of the discussion.

Validity of the findings

As mentioned before, sampling dates are not listed in the study. The authors make the point that the microbiome composition in E. pallida is very variable, if the samples are obtained at different seasons this could add to the variability independently of location; maybe the variation would be lower were all samples collected in the same season and / or year.
Similarly, and information on sampling depth is missing. This should also be added if available.

As also mentioned before, I find the use of ethanol preserved museum specimens for microbiome analysis a little worrying. Ideally there would be an experiment how fresh and preserved ethanol samples change over time with these methods, maybe this exists in the literature. In any case this issue needs to be discussed in the manuscript, ideally with reference to previous experiments, as it could lead to a strong bias.

Line 243: It seems to be prevalent in the coral community to assume that all and every bacterial species that form a stable association is a beneficial symbiont. Of course pathogens and parasites are also often very specific to their host and can be found in a large percentage of individuals of most species, without functional data it is hard to tell these groups apart. Please change this accordingly.

Line 253ff.: I’m not sure what this sentence means, the study is descriptive and does not look at the “interplay” between host and symbiont diversity, especially not functionally.

Additional comments

Line 69: Why would the microbial diversity be correlated with the complexity of the host body plan? Sponges have a simple body plan and a very high diversity of bacterial symbionts.

Minor corrections:
L143: OTU -> OTUs
L451: V1-V2 –> V1-V4

---

## Round 0.2 · accepted · Accept

The manuscript has been well revised and is now acceptable for publication. Please note I have made a few small edits; these can be incorporated at the proof stage or earlier and are in the attached PDF file.

I look forward to seeing the published version of your work.

Reviewer 2 ·

Basic reporting

This article has met the basic requirements of this journal.

Experimental design

In this version, authors improved their description of methods. The revision of Symbiodinium part strengthened the investigation.

Validity of the findings

no comment

Additional comments

The authors replied the rebuttal clearly, and revised the manuscript well. I think this article is ready to publish.

Reviewer 3 ·

Basic reporting

The methods section has been expanded and now gives sufficient information.

Experimental design

The issues were addressed in the rebuttal letter.

Validity of the findings

I still think ethanol preserved samples are not ideal for this type of study, but with the added sampling information the reader can be the judge of this.

Additional comments

All concerns have been addressed satisfactorily and I would deem the article fit for publication.